**Data Availability Statement:** All data are available from the Open Science Framework (OSF). doi: 10.17605/OSF.IO/6D8UX.

**Funding:** This work is supported by the Netherlands Organization for Health Research and

# Who accepts nudges? nudge acceptability from a self-regulation perspective

**Laurens C. van Gestel**◉*, Marieke A. Adriaanse, Denise T. D. de Ridder

Department of Social, Health and Organizational Psychology, Utrecht University, Utrecht, Netherlands

* l.c.vangestel@uu.nl

## Abstract

### Background

Public acceptability of nudging is receiving increasingly more attention, but studies remain limited to evaluations of aspects of the nudge itself or (inferred intentions) of the nudger. Yet, it is important to investigate which individuals are likely to accept nudges, as those who are supposed to benefit from the implementation should not oppose it. The main objective of this study was to integrate research on self-regulation and nudging, and to examine acceptability of nudges as a function of self-regulation capacity and motivation.

### Method

Participants ($N = 301$) filled in questionnaires about several components of self-regulation capacity (self-control, proactive coping competence, self-efficacy, perceived control and perceived difficulty) and motivation (autonomous motivation and controlled motivation). To evaluate nudge acceptability, we used three vignettes describing three types of nudges (default, portion size, and rearrangement) that stimulated either a pro-self behavior (healthy eating) or pro-social behavior (sustainable eating) and asked participants to rate the nudges on (aspects of) acceptability.

### Results

Results revealed that there were substantial differences in acceptability between the three types of nudges, such that the default nudge was seen as less acceptable and the rearrangement nudge as most acceptable. The behavior that was stimulated did not affect acceptability, even though the nudges that targeted healthy eating were seen as more pro-self than the nudges targeting sustainable eating. From all self-regulation components, autonomous motivation was the only measure that was consistently associated with nudge acceptability across the three nudges. For self-regulatory capacity, only some elements were occasionally related to acceptability for some nudges.

Development under ZonMW Grant 91215012
(www.zonmw.nl). The funders had no role in study
design, data collection and analysis, decision to
publish, or preparation of the manuscript.

## Conclusion

The current study thus shows that people are more inclined to accept nudges that target behaviors that they are autonomously motivated for, while people do not meaningfully base their judgments of acceptability on self-regulatory capacity.

## Introduction

For years, behavioral science has focused on improving self-regulation as a central route for success in life [1]. More recently, the focus of behavior change experts gradually shifted from the individual and its capacities to the environment in which the individual navigates. With this shift came the interest in nudging as a novel and supplemental behavior change technique [2]. Yet, currently there is little knowledge about the interplay between self-regulatory processes and strategic changes in the environment to stimulate desirable behavior, and there is a need for integration of these two routes towards behavior change in order to obtain a better understanding of the promise and pitfalls of nudging. Research on a more detailed understanding of when and for whom nudges are effective is emerging [3–5], but similarly, one could expect that people who are more motivated and/or more capable to self-regulate could differ in the extent to which they are open to being nudged. For example, one could speculate that those who are successful at self-regulating could be the ones who welcome nudges as they do not feel threatened themselves, but one could similarly expect that those who are less successful in self-regulating might welcome nudges as they recognize its potential in helping them achieve their desired end states. In the current study we investigate the link between self-regulation and openness to being nudged through the lens of nudge acceptability.

Currently, there is a lack of knowledge about which individuals are more prone to accept nudges than others. It is important to get a better understanding of this, given that the very people who are supposed to benefit from a nudge intervention should welcome, or at the very least not oppose, the implementation of the nudge. We thus propose to focus more on individual self-regulation capacity and motivation in studying nudges' acceptability and aim to shed light on the question whether those who need it and/or those who want it have more favorable views about nudges. In doing so, we conducted a study on the relation between several important self-regulation concepts and nudges' acceptability across a pro-self (healthy eating) and a pro-social (sustainable eating) behavior. This distinction between pro-self and pro-social nudges has–apart from differing implications for self-regulation capacity and motivation–previously led to differences in judgments of acceptability, such that pro-social nudges that contributed to the greater good were judged as less acceptable [6].

### Nudge acceptability

Public acceptability is a core element for the systematic uptake of evidence-based practices in real-world contexts [7], and a lack of acceptability can often pose a barrier to successful implementation of novel interventions [8, 9]. Population wide statistics regarding the acceptance of nudging are generally high and generally demonstrate majority support for the most well-known nudges [6, 10, 11]. So far, research on acceptability of nudges has mostly focused on factors inherent to the nudge, such as actual, perceived, and communicated effectiveness of the nudge [8, 12–15], the target group of the nudge [12], and perceived intrusiveness of the nudge [16]. Recent studies have also focused on nudges' acceptability in relation to dispositions and

intentions of the policy maker implementing the nudge, such as the source of the nudge [12], his or her political orientation [17], and trustworthiness of the nudger [16]. Taken together, there is a growing number of studies that focus on the acceptability of nudges, but, as illustrated above, the focus remains largely limited to aspects of the nudge or nudger.

Individual factors that reveal which people are more likely to accept nudges, however, have rarely been studied, and studies thus far focused on largely stable factors such as traits and demographics [8, 18]. These studies have mostly revealed no or inconsistent associations between those factors and nudge acceptability. Yet, nudges' acceptability may possibly also depend on self-regulatory capacity–that determines whether people are able to adjust their behavior in line with their goals–and motivation. From an implementation science perspective one ought to know whether nudges are accepted by those who are being targeted by the intervention, while from an integrative behavioral science perspective more knowledge is required about the interplay between individual motivation and capacity and interventions in the immediate choice environment. In other words, a thorough investigation of individual factors that move beyond demographics is crucially missing, and public policy makers should be better informed about whether their target group that should benefit from the nudge is also likely to accept it.

## Self-regulation and nudge acceptability

When investigating nudge acceptability from a self-regulation perspective, one could identify a plethora of potentially relevant factors. Inspired by two related frameworks for understanding human self-regulatory behavior–the COM-B system which includes capacity, opportunity and motivation [19] and the Fogg Behavior Model which includes ability, prompts, and motivation [20]–we distinguished between two main clusters: capacity and motivation. The nudges themselves can be seen as a prompt or opportunity in these behavioral models.

**Capacity.** Self-control is the ability to transcend short-term gratifications in order to achieve long-term goals [21]. High trait self-control has been associated with advantageous outcomes in several domains such as school and work performance, social relationships and health [22]. Traditionally, self-control has been viewed as the effortful inhibition of unwanted impulses [23]. However, more recently, the notion of effortless self-control has been introduced [24], which denotes that successful self-control relies on rather effortless strategies such as proactively avoiding response conflicts. Related concepts such as situational self-control [25, 26] and self-nudging [27] also highlight the potential of changing one's environment in order to achieve self-control successes. As a consequence, nudges may be appreciated as helpful in acting upon self-control, as nudges may take away potential barriers to achieve long-term success. On the contrary, nudges may also be seen as intrusive or irrelevant, and may invoke feelings of reactance [28], especially among those who consider themselves high in self-control. Nevertheless, recent developments in self-control research highlight its potential importance in predicting nudge acceptability, and give reason to suspect that successful self-control and nudging acceptability may be related.

Apart from having low or high self-control, individuals may differ in the extent to which they possess a rich toolbox of self-regulatory skills. We specifically focus on a set of skills that are referred to as proactive coping skills: "efforts undertaken in advance of a potentially stressful event to prevent it or to modify its form before it occurs" [29, p. 417]. This set of skills consists of actions that one can take *prior to* exposure to a potential stressor, and it includes skills such as planning and monitoring. Those who are high in proactive coping competence may appreciate a nudge as it may take away an anticipated barrier to achieving one's goals. Finally, we also focused on self-efficacy, which refers to the belief that one can successfully pursue a

course of action in light of potential setbacks [30], as well as perceived control and perceived difficulty of performing the desired behavior.

**Motivation.** Surprisingly little attention has been paid to the role of motivation in nudging acceptability. A common finding in the literature on public attitudes towards policy measures is that people are motivated by self-interest [14, 31]. This implies that those who are likely to benefit from a certain policy measure are most likely to support the policy, while those who will not benefit or will be disadvantaged by a measure are most likely to dislike the policy. This finding has, for example, been found among smokers who are more likely to oppose measures aimed at reducing smoking [14]. However, this typical self-interest finding has thus far received little attention in nudging research. In line with previous research on other types of policies, it is likely that those how are motivated to perform a certain behavior are more likely to support nudges that stimulate that behavior. In our own research, we previously found that acceptability of a default correlated weakly but positively with autonomous motivation and negatively with amotivation, while no association was found with controlled motivation [5]. Yet, apart from our own work, we are currently not aware of any other studies that focused on motivation and nudge acceptability. We intend to build upon our earlier findings and will also focus on autonomous motivation, controlled motivation, and amotivation [32] in the present study.

## The current study

To gain a better understanding of the relation between self-regulation and nudge acceptability, we conducted a study with three vignettes describing three types of nudges. We administered several questionnaires about self-regulation capacity and motivation, related to either healthy food intake or sustainable food choices. Next, depending on the experimental condition, participants read three vignettes with three different nudges that promoted either healthy food choices or sustainable food choices at work and rated the three nudges on our primary variable of interest (acceptability) and three related measures (intrusiveness, perceived effectiveness, and goal alignment). We used the same three vignettes in the healthy eating condition as in the sustainable eating condition, but manipulated the rationale for implementing the nudge such that it would either be seen as a pro-self behavior (healthy eating) or a pro-social behavior (sustainable eating). We did this in order to be able to generalize our results, as previous studies have revealed differences in acceptability dependent on the pro-self or pro-social nature of the nudge [6]. Moreover, these different dimensions could have implications for the understanding of self-regulation. While healthy eating has a rather distant benefit for individual health, sustainable eating possibly has an even more distant benefit that transcends individual gratification. Finally, healthy and sustainable behavior are the two most often used behaviors in nudging research [33]. We did not formulate a priori hypotheses, but did anticipate to find associations between nudge acceptability and factors of self-regulation capacity and motivation. We preregistered the study at As Predicted where we included a basic analysis plan (https://aspredicted.org/blind.php?x=hx2y8a).

## Method

### Participants and design

For half of the participants the study focused on the behavioral domain of healthy eating while for the other half the study focused on sustainable eating. The present study thus used a mixed design with the behavioral domain (healthy eating vs. sustainable eating) as between-subjects factor and type of nudge (default vs. portion size vs. rearrangement) as within-subjects factor. We decided a priori to collect data from 300 participants (150 per behavioral domain), which we deemed adequate to detect a medium-sized effect with 80% power, and substantial enough

to explore the data with enough flexibility. To illustrate, with 150 participants one can detect correlations of at least $r = .20$ with $\alpha = .05$ and $\beta = .80$.

We collected data on Prolific Academic from adult participants with a UK nationality and a minimum approval rate of 95%. We included 301 participants (157 female, 142 male, 2 Other/ Rather not specify; $M_{age} = 38.37$, $SD_{age} = 14.58$). None of the participants failed the two attention checks and thus no participants were excluded from further analyses. Participants were rewarded with £1.00 for their participation. The study was approved by the Ethics Committee of the Faculty of Social and Behavioral Sciences of Utrecht University under number 20–579. We obtained written informed consent from all participants.

## Procedure

Participants were invited to participate in a questionnaire study on self-regulation, motivation, and the environment in which people make food-related decisions. After they provided informed consent, participants were randomly allocated to either the healthy eating or sustainable eating condition. Next, we administered several questionnaires. The first set of questionnaires pertained to self-regulatory concepts related to healthy eating or sustainable eating. We then asked participants to rate the trustworthiness of their employer. Subsequently, in order to check our assumption that healthy eating would be seen as more pro-self than sustainable eating, we asked participants to rate the behavior of interest on a dimension of pro-self vs. prosocial. After these questionnaires, we showed participants three vignettes describing three different types of nudges in random order.

The *vignettes* described a situation after the COVID-19 pandemic in which the participant's employer had decided to promote good health or sustainability via a nudging intervention. For each vignette, participants were asked to rate acceptability of the nudge and related concepts, which served as our dependent measures. After the questions about the nudges, we asked participants for their demographics (age and gender), for their frequency of going to work, for their frequency of buying food at work, and ended with an open question in which participants could write any final thoughts. Finally, participants were debriefed, thanked and paid for their participation.

## Measures and materials

**Self-regulatory capacity.** Self-regulatory capacity was assessed in three parts: (1) Self-control, (2) proactive coping competence, and (3) self-efficacy, perceived control, and perceived difficulty.

*Self-control.* Self-control was measured with the Brief Self-Control Scale (BSCS) [34]. The scale contains 13 items (e.g., "I am good at resisting temptations") measured on a 5-point Likert scale ranging from 1 (not at all) to 5 (very much). The scale has been validated and applied in previous research [34]. Nine items were reversed before creating the composite score and the scale had good reliability (*Cronbach's* $\alpha = .87$).

*Proactive coping competence.* Proactive coping was measured using the Utrecht Proactive Coping Competence scale (UPCC) [35]. The scale consists of 21 items measured on a 4-point Likert scale ranging from 1 (not competent) to 4 (very competent). Participants were asked to rate their competency of several skills such as "Recognizing signals that something might go wrong", "Translating my desires into plans", and "Evaluating whether I accomplished the goal I wanted to reach". The scale has been validated and applied in previous research [35]. The scale had good reliability (*Cronbach's* $\alpha = .89$).

*Self-efficacy, perceived control, and perceived difficulty.* Self-efficacy ("I am confident in my ability to eat a healthy/sustainable diet"), perceived control ("Eating a healthy/sustainable diet

is in my own hands") and perceived difficulty ("I find it difficult to eat a healthy/sustainable diet") were measured with one item each on a 7-point Likert scale ranging from 1 (not at all) to 7 (very much).

**Motivation.** In line with Van Gestel and colleagues [5], we measured three different types of motivation: Autonomous motivation, controlled motivation, and amotivation. These different types of motivation for eating a healthy or sustainable diet were measured using the Treatment Self-Regulation Questionnaire (TSRQ) [36]. Participants were asked to rate reasons for eating a healthy or sustainable diet, dependent on the condition they were in. The scale consists of 15 items: 6 for autonomous motivation (e.g., "Because I feel that I want to take responsibility for my own health"), 6 for controlled motivation (e.g., "Because I feel pressure from others to do so"), and 3 for amotivation (e.g., "I really don't think about it"). The scale has been validated and used in previous research for both behaviors of interest [5, 34]. The subscales for autonomous motivation (*Cronbach's α* = .90) and controlled motivation (*Cronbach's α* = .82) showed good reliability. The subscale for amotivation (*Cronbach's α* = .58) had poor reliability and was excluded from further analyses.

**Trustworthiness.** Trustworthiness of the employer was measured with 1 item ("To what degree do you regard your employer as trustworthy?") on a 7-point Likert scale ranging from 1 (not at all trustworthy) to 7 (very trustworthy).

**Pro-self vs. pro-social dimension.** To assess our assumption that eating a healthy diet is more of a pro-self behavior than eating a sustainable diet, we asked participants to rate one statement ("Eating a healthy/sustainable diet is something I do for. . .") on a continuous slider ranging from 0 (myself) to 100 (society).

**Vignettes.** We used three vignettes for each behavioral domain that described three different nudges: default, portion size, and rearrangement. We chose these three behaviorally-oriented nudges as these have been shown to be among the most effective in the domain of eating behavior [37]. The vignettes described a situation in which the employer would encourage healthy or sustainable eating via one of these nudging interventions. The formulation of the vignettes largely followed the same structure: "In order to promote [good health/sustainability] by [behavioral outcome], your employer has decided that [nudge]." See Table 1 for a full description of all six nudge vignettes.

**Ratings of nudges.** For each vignette, we asked participants to rate several aspects of the nudge. The main dependent variable–the extent to which people would accept the nudges–was measured with three items ("How much would you accept the implementation of this measure?", "How much would you appreciate the implementation of this measure?", and "How much would you support the implementation of this measure?"), all measured on a continuous slider ranging from 0% to 100%. These three items were averaged into a composite score for acceptability of the nudge (*Cronbach's α* = .94). Other measures pertained to ratings of

**Table 1. The nudge vignettes used in the healthy eating or sustainable eating conditions describing the default nudge, portion size nudge, or rearrangement nudge.**

| | Healthy eating | Sustainable eating |
|---|---|---|
| Default | In order to promote good health by eating less meat, your employer has decided that all lunch orders are now automatically vegetarian, unless otherwise specified. | In order to promote sustainability by eating less meat, your employer has decided that all lunch orders are now automatically vegetarian, unless otherwise specified. |
| Portion size | In order to promote good health by reducing portion sizes, your employer has decided to use smaller plates to reduce consumption in the self-service cafeteria. | In order to promote sustainability by reducing portion sizes, your employer has decided to use smaller plates to reduce food waste in the self-service cafeteria. |
| Rearrangement | In order to promote good health by eating differently, your employer has decided to rearrange the buffet such that healthier foods are presented first. | In order to promote sustainability by eating differently, your employer has decided to rearrange the buffet such that more sustainable foods are presented first. |

intrusiveness ("How intrusive do you find this measure?"), perceived effectiveness ("How effective do you think this measure would be?"), and goal alignment ("To what extent is this measure in line with your own goal?"). These were measured with one item each, also on a continuous slider ranging from 0% to 100%.

**Demographics.** We asked participants for their age (in years) and gender (female, male, other/rather not specify). We asked participants for the frequency of going to work ("In the past two weeks, how often did you go to work?") with answer options ranging from 0 to 14 times in total, plus an option to indicate unemployment. We also asked for the frequency of buying food at work ("How often do you buy something to eat at work?") with the answer options never, rarely, sometimes, often, always, and not employed. Finally, we included one open question in which participants could write anything they deemed relevant.

## Results

Data and code are available on the Open Science Framework (https://osf.io/cnsdm/).

### Preprocessing steps

As preregistered, outliers were defined as 3 *SD*s away from the mean and were set missing. This only applied to the measure of perceived control (5 participants). All analyses were run with inclusion and exclusion of these outliers, but this did not change any of the results. Therefore, we report on the entire sample with inclusion of outliers.

### Descriptives

On average, participants regarded their employer as quite trustworthy (*M* = 4.90, *SD* = 1.54). At the time of data collection (December 2020), about a third of the participants worked from home completely (*N* = 109). 62 participants were unemployed. The remaining 130 participants on average had gone to their work location 6.27 (*SD* = 3.74) times in the previous two weeks. Given the relatively high number of unemployed participants, we also ran the main analyses with the subsample of employed participants. The overall pattern of results was consistent with the results reported for the entire sample, although there were some minor differences in significance of certain predictor variables. In the results section we report on the entire sample, but we include the results for the subsample in the Supplementary Online Materials.

Participants were at least somewhat familiar with buying food items at work. Of those who had indicated to be employed, 31% often or always bought food at work, 23% sometimes did so, and finally 46% rarely or never did so.

Participants on average scored around the mid-point of the self-control scale (*M* = 3.00, *SD* = .70) and reported to be relatively competent in proactive coping (*M* = 2.91, *SD* = .41). Participants also reported to feel relatively efficacious about eating a healthy or sustainable diet (*M* = 4.67, *SD* = 1.51), felt in control (*M* = 5.76, *SD* = 1.19), and scored around the midpoint of the scale for perceived difficulty (*M* = 4.11, *SD* = 1.64). Motivation for autonomous reasons was relatively high (*M* = 4.86, *SD* = 1.29), while motivation for controlled reasons was considerably lower (*M* = 3.04, *SD* = 1.22). Full descriptives and correlation coefficients of the self-regulatory concepts are reported in Table 2.

### Ratings of nudges

The nudges that targeted healthy eating were evaluated as more pro-self (*M* = 15.93, *SD* = 17.82) than the nudges that targeted sustainable eating (*M* = 47.97, *SD* = 25.73), *t*(273) = -12.61, *p* < .001, *d* = 1.44, thereby confirming our underlying assumption that healthy eating

**Table 2. Descriptives and correlation coefficients for self-regulatory concepts.**

| | Mean (SD) | Range | 1 | 2 | 3 | 4 | 5 | 6 | 7 | 8 |
|---|---|---|---|---|---|---|---|---|---|---|
| 1. Self-Control | 3.00 (0.70) | 1.23–5.00 | (.87) | | | | | | | |
| 2. Proactive Coping Competence | 2.91 (0.43) | 1.71–4.00 | .49*** | (.89) | | | | | | |
| 3. Self-efficacy | 4.67 (1.51) | 1.00–7.00 | .15** | .14* | | | | | | |
| 4. Perceived Control | 5.76 (1.19) | 1.00–7.00 | .00 | .17** | .32*** | | | | | |
| 5. Perceived Difficulty | 4.11 (1.64) | 1.00–7.00 | -.19*** | -.06 | -.59*** | -.24*** | | | | |
| 6. Autonomous Motivation | 4.86 (1.29) | 1.00–7.00 | .03 | .11 | .50*** | .15* | -.31*** | (.90) | | |
| 7. Controlled Motivation | 3.04 (1.22) | 1.00–7.00 | -.25*** | -.15* | .24*** | .00 | -.05 | .35*** | (.82) | |
| 8. Amotivation | 2.88 (1.27) | 1.00–6.00 | -.11 | -.14* | -.27*** | .04 | .29*** | -.51*** | .14* | (.58) |

*Note.* Cronbach's alphas are shown in the diagonal.

\*\*\* $p < .001$,

\*\* $p < .01$,

\* $p < .05$

would be seen as more of a pro-self behavior than sustainable eating. Descriptives and correlation coefficients for the ratings of all three nudges are reported in Table 3.

Before investigating the relation between self-regulation and nudge acceptability, we first analyzed acceptability of the nudges as a function of the type of nudge and behavioral domain.

**Acceptability.** We found a large effect of the type of nudge on ratings of acceptability, $F(2, 598) = 105.42$, $p < .001$, $\eta_p^2 = .26$. Post-hoc comparisons using Bonferroni adjustment revealed that the rearrangement nudge ($M = 70.81$, $SD = 25.64$) was evaluated as significantly more acceptable than the portion size nudge ($M = 50.45$, $SD = 30.69$, $p_{adj} < .001$) which, in turn, was evaluated as significantly more acceptable than the default nudge ($M = 42.74$, $SD = 34.08$, $p_{adj} = .002$). Acceptability of the nudges did not differ by behavioral domain, $F(1, 299) = 1.74$, $p =$

**Table 3. Descriptives and correlation coefficients for the ratings of the three types of nudges.**

| | Mean (SD) | 1 | 2 | 3 | 4 |
|---|---|---|---|---|---|
| **Default** | | | | | |
| 1. Acceptability | 42.74 (34.08) | | | | |
| 2. Intrusiveness | 65.24 (34.00) | -.69*** | | | |
| 3. Effectiveness | 37.79 (28.30) | .62*** | -.51*** | | |
| 4. Alignment | 41.20 (34.39) | .81*** | -.55*** | .53*** | |
| **Portion Size** | | | | | |
| 1. Acceptability | 50.45 (34.08) | | | | |
| 2. Intrusiveness | 46.98 (33.04) | -.52*** | | | |
| 3. Effectiveness | 48.89 (27.78) | .65*** | -.34*** | | |
| 4. Alignment | 47.14 (30.44) | .76*** | -.31*** | .59*** | |
| **Rearrangement** | | | | | |
| 1. Acceptability | 70.81 (25.64) | | | | |
| 2. Intrusiveness | 23.20 (28.26) | -.48*** | | | |
| 3. Effectiveness | 53.92 (26.52) | .51*** | -.11* | | |
| 4. Alignment | 56.18 (27.41) | .65*** | -.17** | .53*** | |

*Note.* \*\*\* $p < .001$,

\*\* $p < .01$,

\* $p < .05$

.188, but we did observe a significant interaction effect between the behavioral domain and type of nudge, $F(2, 598) = 3.45$, $p = .032$, $\eta_p^2 = .01$. This effect was driven by ratings of the portion size nudge, which was evaluated as more acceptable when targeting sustainable eating ($M = 54.76$, $SD = 31.03$) than when targeting healthy eating ($M = 45.93$, $SD = 29.77$, $p_{adj} = .012$). See Fig 1 for a graphical overview of the results for acceptability.

**Intrusiveness, perceived effectiveness, and goal alignment.** For the other three dependent variables that were related to acceptability, we found a similar pattern of results. Most importantly, we consistently found differences between the three types of nudges with medium to large effect sizes (all $ps < .001$). The direction of these effects was consistent with the overall pattern of nudge acceptability, such that the rearrangement nudge was evaluated as less intrusive, more effective, and more in line with personal goals than the portion size nudge, which in

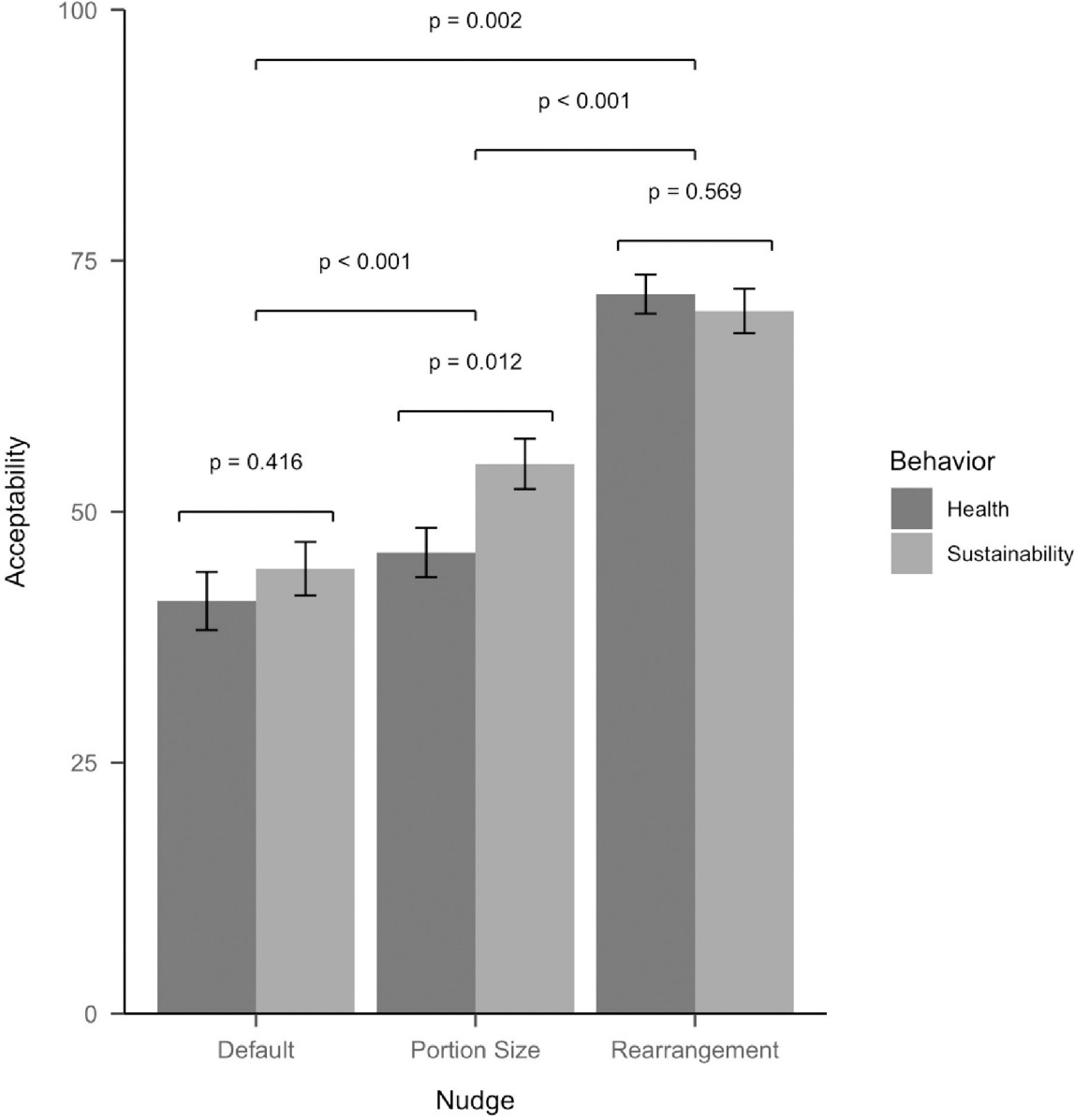

**Fig 1. Mean scores of acceptability of the three nudges by behavioral domain.** Error bars represent standard errors.

turn was evaluated as less intrusive, more effective, and more in line with personal goals than the default nudge. For intrusiveness, we also found a small effect of the behavioral domain and a small interaction effect of the behavioral domain and type of nudge, and for goal alignment we also found a small interaction effect. See Table 4 for a complete overview of the results for all four dependent variables.

## Main analyses

As the main purpose of this study, we performed the main analyses focusing on the relation between self-regulation and nudge acceptability. Taken together, the results above revealed that the rearrangement nudge was evaluated as more acceptable, less intrusive, more effective, and more in line with personal goals than the portion size nudge, which in turn was evaluated as more acceptable, less intrusive, more effective, and more in line with personal goals than the default nudge. Given this consistent pattern with medium to large effect sizes, we considered it warranted to conduct the main analyses separately for each type of nudge. The behavioral domain did not affect ratings of acceptability, and thus we did not distinguish between healthy eating and sustainable eating in our main analyses. Thus, in order to explore the relation between self-regulation and nudge acceptability, for each nudge we regressed acceptability, intrusiveness, perceived effectiveness, and goal alignment on all measured self-regulation constructs.

**Acceptability.** For all three nudges, the linear regression models significantly fitted the data (all $ps < .001$), with model fit ranging from $R^2_{adj} = .07$ to $R^2_{adj} = .22$. Autonomous motivation significantly predicted acceptability of all three nudges: default ($\beta = .29$, $p < .001$), portion size ($\beta = .22$, $p = .001$), and rearrangement nudge ($\beta = .52$, $p < .001$). Self-control ($\beta = .22$, $p = .001$) and self-efficacy ($\beta = -.19$, $p = .014$) also significantly predicted acceptability of the portion size nudge, while perceived control ($\beta = -.14$, $p = .017$) significantly predicted acceptability for the default nudge. Thus, regardless of the type of nudge and behavioral outcome, autonomous motivation was the only measure that was consistently related to nudge acceptability.

**Table 4. Mixed ANOVAs predicting acceptability, intrusiveness, effectiveness and goal alignment.**

|  | F | $df_n, df_d$ | p | $\eta_p^2$ |
|---|---|---|---|---|
| **Acceptability** |  |  |  |  |
| Nudge | 105.42 | (2, 598) | < .001 | .26 |
| Behavior | 1.74 | (1, 299) | .188 | .01 |
| Nudge X Behavior | 3.45 | (2, 598) | .032 | .01 |
| **Intrusiveness** |  |  |  |  |
| Nudge | 184.47 | (2, 598) | < .001 | .38 |
| Behavior | 5.26 | (1, 299) | .023 | .02 |
| Nudge X Behavior | 4.91 | (2, 598) | .008 | .02 |
| **Effectiveness** |  |  |  |  |
| Nudge | 37.69 | (2, 598) | < .001 | .11 |
| Behavior | 0.45 | (1, 299) | .503 | .00 |
| Nudge X Behavior | 0.20 | (2, 598) | .816 | .00 |
| **Alignment** |  |  |  |  |
| Nudge | 27.84 | (2, 598) | < .001 | .09 |
| Behavior | 0.63 | (1, 299) | .427 | .00 |
| Nudge X Behavior | 4.89 | (2, 598) | .008 | .02 |

**Intrusiveness, perceived effectiveness, and goal alignment.** For intrusiveness, autonomous motivation (negatively) predicted ratings for the default nudge and rearrangement nudge. The overall regression model for intrusiveness of the portion size nudge did not reach significance ($p$ = .379) and none of the predictors were significant. For perceived effectiveness, autonomous motivation (positively) significantly predicted ratings for the default and rearrangement nudge, and marginally significantly predicted ratings of the portion size nudge ($p$ = .075). However, the overall regression model for perceived effectiveness of the default nudge was marginally significant ($p$ = .055). For goal alignment, autonomous motivation again significantly (positively) predicted ratings for all three types of nudges. Together, this shows that autonomous motivation was also the most important variable associated with aspects of nudge acceptability.

Most other self-regulation constructs did not significantly predict aspects of nudge acceptability, and this did not reveal a consistent pattern across the ratings of aspects of nudge acceptability. For intrusiveness, perceived control also (positively) predicted ratings for the default nudge, while for perceived effectiveness, self-control also (positively) predicted ratings of the portion size nudge. Finally, for alignment, self-control (positively) and self-efficacy (negatively) also predicted ratings for the portion size nudge, while controlled motivation also (positively) predicted ratings of the rearrangement nudge. See Table 5 for a full overview of the regression results for all dependent variables.

## General discussion

In this study we aimed to investigate the relation between two main components of self-regulation and nudge acceptability across different types of nudges targeting pro-self behavior (healthy eating) and pro-social behavior (sustainable eating). Previous studies generally demonstrated majority support for most nudges [10, 11] and have established evidence for important predictors of nudge acceptability that lie within aspects of the nudge itself [12, 13] or are inferred from (the intentions of) the nudger [17]. Yet, it is pivotal to identify whether the very people who are intended to benefit from the nudge are likely to accept or oppose it. Therefore, with this study we investigate predictors related to self-regulation capacity (self-control, proactive coping competence, self-efficacy, perceived control, and perceived difficulty) and motivation (autonomous and controlled) as possible determinants of nudge acceptability. In doing so, we also aim to bring together two routes to behavior change (through improving self-regulation or changing the environment) which to date have mainly been studied separately.

The current study showed that only autonomous motivation was consistently related to (aspects of) acceptability across the three types of nudges. This finding is in line with a previous study that showed a correlation between autonomous motivation and acceptability of a default nudge [5] and provides first evidence for the notion that those who are motivated to perform a certain type of behavior are more likely to embrace a nudge that aims to stimulate that behavior as an acceptable policy instrument. This study thus complements previous work that focused on the role of autonomous motivation for the effectiveness of nudges [5], and on the need for autonomy as a consequence of being nudged [38–40]. As autonomous motivated behavior is performed out of interest or enjoyment [32], the current findings indicate that motivation for personally endorsed reasons is positively related to nudge acceptability. This relation was consistent across pro-self and pro-social nudges, suggesting that nudge acceptability for autonomously motivated reasons transcends beyond purely individual benefits and also includes behavior that holds advantages for society at large. In a way, this study thus shows that acceptability of nudges is not purely limited to aspects of self-interest [14, 31]. Controlled motivation was not related to acceptability of any of the three nudges.

**Table 5. Linear regression models predicting acceptability, intrusiveness, perceived effectiveness and goal alignment for the three different types of nudges.**

| | Acceptability | Intrusiveness | Effectiveness | Alignment |
|---|---|---|---|---|
| | $\beta$ (SE) | $\beta$ (SE) | $\beta$ (SE) | $\beta$ (SE) |
| **Default** | | | | |
| Self-control | .06 (.07) | .03 (.07) | .06 (.07) | .08 (.06) |
| Proactive Coping | -.04 (.06) | -.02 (.07) | -.02 (.07) | -.07 (.06) |
| Self-efficacy | .10 (.08) | -.02 (.08) | .03 (.08) | .04 (.08) |
| Perceived Control | -.14 (.06) * | .15 (.06) * | -.04 (.06) | -.09 (.06) |
| Perceived Difficulty | -.04 (.07) | .06 (.07) | -.06 (.07) | -.11 (.07) |
| Autonomous Motivation | .29 (.07) *** | -.20 (.07) ** | .16 (.07) * | .34 (.07) *** |
| Controlled Motivation | .00 (.06) | -.02 (.06) | .01 (.06) | .02 (.06) |
| **Portion Size** | | | | |
| Self-control | .22 (.07) ** | -.12 (.07) | .19 (.07) ** | .23 (.07) *** |
| Proactive Coping | .01 (.07) | -.06 (.07) | -.01 (.07) | -.03 (.06) |
| Self-efficacy | -.19 (.08) * | .05 (.08) | -.04 (.08) | -.21 (.08) ** |
| Perceived Control | .03 (.06) | -.04 (.06) | .07 (.06) | .06 (.06) |
| Perceived Difficulty | .06 (.07) | -.03 (.07) | .07 (.07) | .07 (.07) |
| Autonomous Motivation | .22 (.07) ** | -.03 (.07) | .13 (.07) | .29 (.07) *** |
| Controlled Motivation | .04 (.06) | -.06 (.07) | -.01 (.06) | .12 (.06) |
| **Rearrangement** | | | | |
| Self-control | .00 (.06) | -.01 (.07) | -.01 (.07) | .01 (.06) |
| Proactive Coping | .07 (.06) | .05 (.07) | .07 (.07) | -.01 (.06) |
| Self-efficacy | -.02 (.07) | .15 (.08) | -.01 (.08) | .02 (.07) |
| Perceived Control | -.06 (.06) | .06 (.06) | .09 (.06) | -.03 (.05) |
| Perceived Difficulty | .06 (.06) | .09 (.07) | -.07 (.07) | .02 (.06) |
| Autonomous Motivation | .52 (.06) *** | -.35 (.07) *** | .19 (.07) ** | .51 (.06) *** |
| Controlled Motivation | -.03 (.06) | .06 (.06) | .07 (.06) | .12 (.05) * |

*Note*. Model fit for Acceptability: $R^2_{adj}$ = .11 *** (Default); $R^2_{adj}$ = .07 *** (Portion size); $R^2_{adj}$ = .22 *** (Rearrangement). Model fit for Intrusiveness: $R^2_{adj}$ = .04 * (Default); $R^2_{adj}$ = .00 (Portion size); $R^2_{adj}$ = .07 *** (Rearrangement). Model fit for Effectiveness: $R^2_{adj}$ = .02 (Default); $R^2_{adj}$ = .03 * (Portion size); $R^2_{adj}$ = .06 *** (Rearrangement). Model fit for Alignment: $R^2_{adj}$ = .15 *** (Default); $R^2_{adj}$ = .11 *** (Portion size); $R^2_{adj}$ = .30 *** (Rearrangement).

The capacity to self-regulate was only occasionally related to (aspects of) nudge acceptability. We suspected that self-regulatory capacity might predict nudge acceptability, as recent developments in the study of self-control have established the importance of embracing situational factors in achieving self-control success [24–27]. In order to investigate this possible association, we included a wide variety of measures for self-regulation capacity. While proactive coping competence was not at all related to (aspects of) nudge acceptability, trait self-control was only associated with acceptability of the portion size nudge. Similarly, perceived difficulty was not at all related to (aspects of) nudge acceptability, while self-efficacy and perceived control were only once associated with acceptability (of the portion size and default nudge respectively). A potential reason for the rare occurrence of associations between these components of self-regulation capacity and nudge acceptability could be that not everyone possesses insight in their ability to self-regulate. People sometimes tend to overestimate their own self-control [41] and therefore may become more reserved about receiving aid from policy makers [42]. Differences in the extent to which people accurately assess their own ability to self-regulate, and in the extent to which people are open to being confronted with this insight, may thus have confounded the expected effects. Still, the current study shows that it is not likely that those who could need support in achieving their desired end states are those who

are prone to accept nudges. Nor does the study provide evidence for the notion that those who are high in self-control embrace environmental interventions like nudges. Rather, the current study shows that especially those who want to perform the nudged behavior for autonomous reasons are more likely to accept those nudges.

We did not find meaningful differences in (aspects of) acceptability as a result of the behavioral domain, even though the target behavior of healthy eating was evaluated as more pro-self than the target behavior of sustainable eating. Healthy eating was seen as clearly pro-self, while sustainable eating was on average evaluated at the midpoint of the scale ranging from pro-self to pro-social. Thus, even though the nudges targeting sustainable eating were not decisively seen as pro-social, our results seem to contradict previous results that suggested that pro-social nudges are seen as less acceptable than pro-self nudges [6]. In previous studies, different types of nudges were classified as pro-self or pro-social, while in our studies we used the exact same nudge, while manipulating the pro-self or pro-social nature of the nudge. Thus, while we used the same nudges and found no differences by the prosocial nature of the nudge, previous findings may have been confounded by external factors like differences in the types of nudges.

Results also revealed that the three nudges in these studies did not receive unequivocal support. For example, the default nudge did not receive majority support and only the rearrangement nudge was seen as acceptable by a considerable margin. One possible explanation for this could be that we selected three behavior-oriented nudges, which have been found to be among the most effective [37]. Actual effectiveness–as inferred from meta-analytic evidence–is negatively associated with public acceptability [13] and thus we may have selected nudges that are among the least supported. Moreover, so-called System 1 nudges like defaults and portion size nudges are generally less accepted than so-called System 2 nudges like disclosure of information [43]. The differences in acceptability between the three types of nudges in the present study were substantial, but our study is not the first to reveal lower ratings of acceptability of default nudges in comparison with other types of nudges [10]. Furthermore, the default nudge was evaluated as the most intrusive and least effective, and these factors have been associated with lower support for nudges [13, 16].

## Limitations and future research

The current study was the first to relate self-regulation to nudge acceptability and was exploratory in nature. This study opens up possibilities for future research but also has its limitations in establishing causal pathways. Confirmatory research with a pre-registered analysis plan and hypotheses would be required to further establish the robustness of these findings. In this light, both replicating the mostly non-significant effects of capacity as well as further analyzing the role of motivation in nudge acceptability could enhance our understanding of which people are most likely to accept nudges and for what reasons. Obviously, conceptual replications with different methods and operationalizations could be of significant value for answering these questions. We specifically note the limitation that the independent variables were all measured in a cross-sectional survey. Although we attempted to present the study in a neutral fashion without uncovering the goal of the study, we cannot completely rule out the possibility of demand characteristics or social desirability biasing the results [44, 45]. The current study does thus not allow for establishing causality, and future studies should build on the current results and experimentally manipulate relevant variables associated with nudge acceptability. Specifically, the role of autonomous motivation deserves attention in future studies in order to establish whether autonomous motivation *causes* higher rates of acceptability.

Another limitation of the current study was that we asked participants to rate fictitious nudges in a vignette study. Although this is currently the most frequently used method for gaining a better

understanding of nudge acceptability [10, 11], we signal a need for more (field) studies with actual implementation of nudges in the real world [46, 47]. This is all the more important given that there can be a discrepancy in subjective evaluations of nudges between hypothetical situations and actually implemented nudges [40]. In line with this, individuals that had been exposed to a default for a meat-free lunch approved the nudge by a large majority (90%) [48], while comparable meat consumption limiting nudges were only approved by a small majority in fictitious scenarios (52% in the UK sample) [10]. Field experiments should thus not only continue to focus on efficacy, but also measure and report on public acceptability of nudges implemented in real life. Future work should continue to integrate the two routes to behavior change, also in terms of effectiveness, in order to further establish when and for whom nudges are effective and acceptable.

Finally, we note that over half of the participants in our sample were unemployed or worked from home, and that almost half of the participants hardly ever purchased food at work, thereby limiting the personal relevance of the vignettes to the participants. We also note that, on average, participants were generally motivated, at least for autonomous reasons, to make healthy and sustainable food decisions.

## Conclusion

In this study we consistently found support for the relation between autonomous motivation and (aspects of) acceptability of three different nudges: default, portion size and rearrangement. The pattern of results was not affected by the type of behavior that was targeted, even though healthy eating was seen as more pro-self. Altogether, the models including self-regulation capacity and motivation explained up to 22% in variability in acceptability of the nudges. Autonomous motivation was the only measure that was consistently related with acceptability across the three types of nudges. Despite having included a wide variety of measures of self-regulatory capacity, we only found incidental effects of some aspects of self-regulatory capacity for some nudges. Together this suggests that people do not meaningfully base their judgments of acceptability on elements of self-regulation capacity, but rather on their own autonomous desire to perform the behavior of interest. The current study highlights the importance of addressing individual traits and states in predicting nudge acceptability in addition to earlier established aspects of the nudge itself or the nudger. Policy makers will benefit from further research into when and for whom nudges can be an acceptable means of stimulating desired behavior, and whether those who can and/or want to perform the behavior of interest are ultimately those who are also likely to accept nudges that promote such behavior.

## Supporting information

**S1 File.**
(DOCX)

## Author Contributions

**Conceptualization:** Laurens C. van Gestel, Marieke A. Adriaanse, Denise T. D. de Ridder.

**Data curation:** Laurens C. van Gestel.

**Formal analysis:** Laurens C. van Gestel.

**Supervision:** Marieke A. Adriaanse, Denise T. D. de Ridder.

**Writing – original draft:** Laurens C. van Gestel.

**Writing – review & editing:** Marieke A. Adriaanse, Denise T. D. de Ridder.

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
