## [Decision Letter · Decision Letter 0]

30 Jun 2021

PONE-D-21-09389

Who accepts nudges? Nudge Acceptability from a Self-regulation Perspective

PLOS ONE

Dear Dr. van Gestel,

Thank you for submitting your manuscript to PLOS ONE. After careful consideration, we feel that it has merit but does not fully meet PLOS ONE’s publication criteria as it currently stands. Therefore, we invite you to submit a revised version of the manuscript that addresses the points raised during the review process.

Both reviewers found merit in your work and thought the topic was both interesting and timely. Reviewer 1 in particular had some concerns about the research design and what this meant for the robustness of the findings. This Reviewer strongly suggested conducting a second study with more objective measures to increase the confidence in the findings. If you decide to conduct such a study I agree that this would provide a stronger case for the conclusions and strengthen the paper considerably,. If you decide to not conduct and additional study, then I think it would be necessary to present a more elaborated Discussion of the limitations of the methods that were used along possible alternative explanations for the findings given these limitations. This key concern along with the other suggestions noted by this Reviewer should be considered in your revision.

We look forward to receiving your revised manuscript.

Kind regards,

Fuschia M. Sirois, PhD

Academic Editor

PLOS ONE

Additional Editor Comments (if provided):

Reviewers' comments:

Reviewer's Responses to Questions

**Comments to the Author**

1. Is the manuscript technically sound, and do the data support the conclusions?

Reviewer #1: Partly

Reviewer #2: Yes

2. Has the statistical analysis been performed appropriately and rigorously? 

Reviewer #1: Yes

Reviewer #2: Yes

3. Have the authors made all data underlying the findings in their manuscript fully available?

Reviewer #1: Yes

Reviewer #2: Yes

4. Is the manuscript presented in an intelligible fashion and written in standard English?

Reviewer #1: Yes

Reviewer #2: Yes

5. Review Comments to the Author

Reviewer #1: This manuscript seeks to understand whether people are likely to accept being nudged (or having their choice architecture shaped so as to promote certain behaviours). As the authors note, they approach this question from a novel perspective that focuses on whether aspects of the individual who is being nudged influence whether they accept being nudged, rather than whether and how the nature of the nudge or agent doing the nudging shapes acceptance.

The authors draw on research on self-regulation to suggest that the individual factors that are likely to influence whether people accept nudges reflect variables like the nature of people’s motivation in the respective context, their capacity for self-regulation etc. This makes intuitive sense and a cross-sectional survey finds that self-report measures of one of these constructs - namely autonomous motivation - are correlated with people’s (hypothetical) judgements about how they would feel being nudged.

1. These findings are interesting, but (as the authors recognise) the empirical evidence is severely limited by the exclusive use of self-report measures and hypothetical vignettes. As a result, this feels like a first study that needs to be supplemented by additional studies. The authors point to the need to consider people’s responses to ‘real’ nudges, but I would add that it would be useful to also manipulate, rather than measure, self-regulatory processes that seem to be important (e.g. autonomous motivation) in experimental designs.

I really hope that the authors respond to this call to conduct further work, as this is a genuinely exciting and innovative perspective on an important issue. It just needs stronger empirical evidence from more robust methods if readers are to have confidence in the findings.

Other, more minor, questions:

2. Do people need to accept nudges for them to be effective? That is to what extent is acceptance just a happy side effect of an effective intervention versus integral to its efficacy? It would be useful to draw readers attention to evidence that acceptance is important (e.g., the positive correlation between measures of acceptance and perceived efficacy of the nudge?)

3. It seems likely that there is a bidirectional relationship between autonomous motivation and (the acceptance and efficacy of) nudging. For example, Wachner et al. find that nudging affects autonomy, the present paper finds that autonomy affects acceptance (and perceived efficacy of) nudging. If the authors agree, then it might be interesting to consider the implications of a bidirectional relationship in the General Discussion.

Reviewer #2: This paper would seem to open up a new area of inquiry that investigates what features of the behavior, the policy, and the person influence the acceptability of public policies. The research offers one of the first tests of public acceptability of nudges as a social/health policy and benefits from an mixed-model experimental design, and multiple indicators of both motivation and capacity for self-regulation. The findings are both interesting and surprising – prosocial behaviors were more acceptable than pro-self behavior and autonomous motivation but not capacity for self-regulation (or the other indicators of motivation) consistency predicted nudge acceptability. The write-up and analyses are of a high standard, and limitations of the research are appropriately acknowledged.

In sum, this is an impressive study and a good candidate for publication. I had nothing in the way of critical commentary or suggestions about how the manuscript might be improved.

6. PLOS authors have the option to publish the peer review history of their article (what does this mean?). If published, this will include your full peer review and any attached files.

Reviewer #1: No

Reviewer #2: No

---

## [Author Response · Author response to Decision Letter 0]

23 Aug 2021

23-08-2021

Dear Dr. Siriois,

We would like to thank you for considering our manuscript entitled ‘Who accepts nudges? Nudge Acceptability from a Self-regulation Perspective’ for publication in PLOS ONE, and for giving us the opportunity to revise our manuscript.

The reviewers made insightful remarks about the strengths and limitations of our work, and we would like to thank the reviewers for that. We believe that the revised manuscript has improved considerably thanks to the constructive remarks of the reviewers.

Below we will provide our reactions to the points that you raised as well as to the reviewers’ comments and will refer to the manuscript where applicable. We will upload a revision with and without tracked changes to the submission portal.

Once again thank you for considering our manuscript.

Kind regards,

Also on behalf of Marieke Adriaanse and Denise de Ridder,

Laurens van Gestel

PONE-D-21-09389

Who accepts nudges? Nudge Acceptability from a Self-regulation Perspective

PLOS ONE

Dear Dr. van Gestel,

Thank you for submitting your manuscript to PLOS ONE. After careful consideration, we feel that it has merit but does not fully meet PLOS ONE’s publication criteria as it currently stands. Therefore, we invite you to submit a revised version of the manuscript that addresses the points raised during the review process.

Both reviewers found merit in your work and thought the topic was both interesting and timely. Reviewer 1 in particular had some concerns about the research design and what this meant for the robustness of the findings. This Reviewer strongly suggested conducting a second study with more objective measures to increase the confidence in the findings. If you decide to conduct such a study I agree that this would provide a stronger case for the conclusions and strengthen the paper considerably,. If you decide to not conduct and additional study, then I think it would be necessary to present a more elaborated Discussion of the limitations of the methods that were used along possible alternative explanations for the findings given these limitations. This key concern along with the other suggestions noted by this Reviewer should be considered in your revision.

We have carefully read and considered the concerns by Reviewer 1 and now pay extra attention to the limitations of the current study. As pointed out in the original manuscript, this study is exploratory in nature, and we believe that it opens up several routes for future research. We decided to elaborate on the possibilities for future research given the limitations of the current study, rather than conduct an additional study, to stick to the nature of the study as reported in the manuscript. Our response to the specific comments by Reviewer 1 can be found below.

We look forward to receiving your revised manuscript.

Kind regards,

Fuschia M. Sirois, PhD

Academic Editor

PLOS ONE

We will ensure that the grant numbers are correct.

We now include the full ethics statement and indicate to have obtained written informed consent (See page 10 of the Revised Manuscript with Track Changes). 

Additional Editor Comments (if provided):

Reviewers' comments:

Reviewer's Responses to Questions

Comments to the Author

1. Is the manuscript technically sound, and do the data support the conclusions?

Reviewer #1: Partly

Reviewer #2: Yes

2. Has the statistical analysis been performed appropriately and rigorously? 

Reviewer #1: Yes

Reviewer #2: Yes

3. Have the authors made all data underlying the findings in their manuscript fully available?

Reviewer #1: Yes

Reviewer #2: Yes

4. Is the manuscript presented in an intelligible fashion and written in standard English?

Reviewer #1: Yes

Reviewer #2: Yes

5. Review Comments to the Author

Reviewer #1: This manuscript seeks to understand whether people are likely to accept being nudged (or having their choice architecture shaped so as to promote certain behaviours). As the authors note, they approach this question from a novel perspective that focuses on whether aspects of the individual who is being nudged influence whether they accept being nudged, rather than whether and how the nature of the nudge or agent doing the nudging shapes acceptance.

The authors draw on research on self-regulation to suggest that the individual factors that are likely to influence whether people accept nudges reflect variables like the nature of people’s motivation in the respective context, their capacity for self-regulation etc. This makes intuitive sense and a cross-sectional survey finds that self-report measures of one of these constructs - namely autonomous motivation - are correlated with people’s (hypothetical) judgements about how they would feel being nudged.

1. These findings are interesting, but (as the authors recognise) the empirical evidence is severely limited by the exclusive use of self-report measures and hypothetical vignettes. As a result, this feels like a first study that needs to be supplemented by additional studies. The authors point to the need to consider people’s responses to ‘real’ nudges, but I would add that it would be useful to also manipulate, rather than measure, self-regulatory processes that seem to be important (e.g. autonomous motivation) in experimental designs.

We agree with the Reviewer that our study is a first study that opens up opportunities for further research. As we pointed out in the original manuscript, one of the major limitations is the hypothetical nature of our study. We agree with the Reviewer that another limitation is that we only measured, instead of manipulated, our independent variables. We already included a short statement about the need for conceptual replications with different methods and operationalizations, but we now elaborate more on this by highlighting the need to manipulate the self-regulatory processes in future studies (See page 22 of the Revised Manuscript without Track Changes). Besides, just to be sure, we double-checked our language in the parts where we interpret the results to prevent implying causality. 

I really hope that the authors respond to this call to conduct further work, as this is a genuinely exciting and innovative perspective on an important issue. It just needs stronger empirical evidence from more robust methods if readers are to have confidence in the findings.

Thank you for your careful reading and constructive comments. 

Other, more minor, questions:

2. Do people need to accept nudges for them to be effective? That is to what extent is acceptance just a happy side effect of an effective intervention versus integral to its efficacy? It would be useful to draw readers attention to evidence that acceptance is important (e.g., the positive correlation between measures of acceptance and perceived efficacy of the nudge?)

We do not think that acceptance is just a happy side effect, but integral to successful implementation of efficacious nudges in the real world. Policymakers ought to be informed about perceived public acceptability and likely base their decision to implement a nudge not only on its effectiveness but also on public support for the intervention. There is indeed a positive correlation between measures of acceptability and perceived effectiveness of nudges, as shown in previous studies (e.g., Cadario & Chandon, 2019; Petrescu, Hollands, Couturier, Ng, & Marteau, 2016) and also in our current study (See Table 3). Yet, actual effectiveness - as inferred from meta-analytic evidence – is inversely related to public acceptability (Cadario & Chandon, 2019). Communicating evidence about actual effectiveness increases public support (Reynolds, Stautz, Pilling, van der Linden, & Marteau, 2020). All in all, the relationship between effectiveness and acceptability is slightly more complicated than one may anticipate at first sight, but low ratings of public acceptability may nevertheless pose a barrier to implementation. We now shortly and more explicitly address the relevance of studying public acceptability in the third paragraph of the manuscript (See page 5 of the Revised Manuscript without Track Changes). 

3. It seems likely that there is a bidirectional relationship between autonomous motivation and (the acceptance and efficacy of) nudging. For example, Wachner et al. find that nudging affects autonomy, the present paper finds that autonomy affects acceptance (and perceived efficacy of) nudging. If the authors agree, then it might be interesting to consider the implications of a bidirectional relationship in the General Discussion.

We think that the reviewer raises a very interesting point here that deserves more attention in future studies, but are hesitant to draw firm conclusions about the possible bidirectional relationship between autonomy/autonomous motivation on the one hand and the acceptance and efficacy of nudging on the other hand based on the evidence that currently exists in the literature. That is, it seems that nudging does not actually have negative consequences for autonomy. While people expect to experience less autonomy when they think about being nudged (Wachner, Adriaanse, & De Ridder, 2020a), they do not report experiencing less autonomy when actually being nudged (Wachner, Adriaanse, & De Ridder, 2020b; Under Review). Thus, autonomy is not necessarily harmed by effective nudges. In another line of research that we conducted previously, we found that autonomous motivation is positively related to performing the desired behavior in the presence of an effective nudge, but that it does not interact with the nudge itself such that the nudge becomes more (or less) effective (Van Gestel, Adriaanse, & De Ridder, 2021). In one of those studies, we found no effect of having been nudged on levels of motivation (Study 2). Finally, the current manuscript describes the positive relationship between autonomous motivation and acceptability of nudges. Taken together, these findings do not currently provide a strong case for the bidirectional relationship, and thus we decided to not speculate about it in the current manuscript. We did, however, decide to shortly refer to other studies that focused on autonomy/autonomous motivation and nudging in order to place the results of the current study in the broader research context (See page 19 of the Revised Manuscript with Track Changes). 

We would like to thank the Reviewer once again for his/her constructive comments and thought-provoking suggestions.

Reviewer #2: This paper would seem to open up a new area of inquiry that investigates what features of the behavior, the policy, and the person influence the acceptability of public policies. The research offers one of the first tests of public acceptability of nudges as a social/health policy and benefits from an mixed-model experimental design, and multiple indicators of both motivation and capacity for self-regulation. The findings are both interesting and surprising – prosocial behaviors were more acceptable than pro-self behavior and autonomous motivation but not capacity for self-regulation (or the other indicators of motivation) consistency predicted nudge acceptability. The write-up and analyses are of a high standard, and limitations of the research are appropriately acknowledged.

In sum, this is an impressive study and a good candidate for publication. I had nothing in the way of critical commentary or suggestions about how the manuscript might be improved.

We thank the reviewer for his/her careful reading and appreciation of our work. 

6. PLOS authors have the option to publish the peer review history of their article (what does this mean?). If published, this will include your full peer review and any attached files.

Do you want your identity to be public for this peer review? For information about this choice, including consent withdrawal, please see our Privacy Policy.

Reviewer #1: No

Reviewer #2: No

References:

Cadario, R., & Chandon, P. (2019). Effectiveness or consumer acceptance? Tradeoffs in selecting healthy eating nudges. Food policy, 85, 1-6.

Petrescu, D. C., Hollands, G. J., Couturier, D. L., Ng, Y. L., & Marteau, T. M. (2016). Public acceptability in the UK and USA of nudging to reduce obesity: the example of reducing sugar-sweetened beverages consumption. PLoS One, 11(6), e0155995.

Reynolds, J. P., Stautz, K., Pilling, M., van der Linden, S., & Marteau, T. M. (2020). Communicating the effectiveness and ineffectiveness of government policies and their impact on public support: a systematic review with meta-analysis. Royal Society Open Science, 7(1), 190522.

Wachner, J., Adriaanse, M. A., & De Ridder, D. T. D. (2020). And how would that make you feel? How people expect nudges to influence their sense of autonomy. Frontiers in Psychology, 11, 3532.

Wachner, J., Adriaanse, M. A., & De Ridder, D. T. D. (2020). The influence of nudge transparency on the experience of autonomy. Comprehensive Results in Social Psychology, 1-15.

Wachner J., Adriaanse M. A., & De Ridder D. T. D. The effect of nudges on autonomy in hypothetical and real life settings. In Press.

Van Gestel, L. C., Adriaanse, M. A., & De Ridder, D. T. (2021). Motivated by default—How nudges facilitate people to act in line with their motivation. Motivation Science. Advance Online Publication.

---

## [Decision Letter · Decision Letter 1]

12 Nov 2021

Who Accepts Nudges? Nudge Acceptability from a Self-regulation Perspective

PONE-D-21-09389R1

Dear Dr. van Gestel,

We’re pleased to inform you that your manuscript has been judged scientifically suitable for publication and will be formally accepted for publication once it meets all outstanding technical requirements.

Kind regards,

Fuschia M. Sirois, PhD

Academic Editor

PLOS ONE

Additional Editor Comments (optional):

Reviewers' comments:

Reviewer's Responses to Questions

**Comments to the Author**

1. If the authors have adequately addressed your comments raised in a previous round of review and you feel that this manuscript is now acceptable for publication, you may indicate that here to bypass the “Comments to the Author” section, enter your conflict of interest statement in the “Confidential to Editor” section, and submit your "Accept" recommendation.

Reviewer #1: (No Response)

Reviewer #2: All comments have been addressed

2. Is the manuscript technically sound, and do the data support the conclusions?

Reviewer #1: Partly

Reviewer #2: Yes

3. Has the statistical analysis been performed appropriately and rigorously? 

Reviewer #1: Yes

Reviewer #2: Yes

4. Have the authors made all data underlying the findings in their manuscript fully available?

Reviewer #1: Yes

Reviewer #2: Yes

5. Is the manuscript presented in an intelligible fashion and written in standard English?

Reviewer #1: Yes

Reviewer #2: Yes

6. Review Comments to the Author

Reviewer #1: As I stated in my original review, this manuscript presents an exciting and innovative perspective on an important issue. However, I was also clear that the authors needed stronger empirical evidence from more robust methods than a single cross-sectional survey, using self-report measures in response to hypothetical scenarios if readers are to have confidence in the findings. My suggestion (echoed by the Editor) was that the authors use these initial findinfs as the basis for further work – e.g., manipulating, rather than measuring, self-regulatory processes in experimental designs.

Sadly, the authors haven’t taken up this challenge, so my evaluation is unchanged. Noting the limitations (and that the study is ‘exploratory’) simply makes explicit that the present evidence is insufficient to draw robust conclusions.

Reviewer #2: As with any new line of inquiry, the present study leaves many questions unanswered. The strength of this paper lies in opening up that line of inquiry and in offering preliminary evidence concerning factors that determine the acceptability of nudges. The findings offered here will need to be corroborated in future research. Importantly, however, this paper should inspire future research and theoretical development -- for that reason, I support publication of this version of the manuscript.

7. PLOS authors have the option to publish the peer review history of their article (what does this mean?). If published, this will include your full peer review and any attached files.

Reviewer #1: No

Reviewer #2: No

---

## [Editor Report · Acceptance letter]

26 Nov 2021

PONE-D-21-09389R1 

Who accepts nudges?
Nudge Acceptability from a Self-regulation Perspective 

Dear Dr. van Gestel:

I'm pleased to inform you that your manuscript has been deemed suitable for publication in PLOS ONE. Congratulations! Your manuscript is now with our production department. 

Kind regards, 

on behalf of

Dr. Fuschia M. Sirois 

Academic Editor

PLOS ONE